# Unlabeled Principal Component Analysis

**Yunzhen Yao, Liangzu Peng, Manolis C. Tsakiris**
School of Information Science and Technology
ShanghaiTech University
`yaoyzh,penglz,mtsakiris@shanghaitech.edu.cn`

## Abstract

We introduce robust principal component analysis from a data matrix in which the entries of its columns have been corrupted by permutations, termed Unlabeled Principal Component Analysis (UPCA). Using algebraic geometry, we establish that UPCA is a well-defined algebraic problem in the sense that the only matrices of minimal rank that agree with the given data are row-permutations of the ground-truth matrix, arising as the unique solutions of a polynomial system of equations. Further, we propose an efficient two-stage algorithmic pipeline for UPCA suitable for the practically relevant case where only a fraction of the data have been permuted. Stage-I employs outlier-robust PCA methods to estimate the ground-truth column-space. Equipped with the column-space, Stage-II applies recent methods for unlabeled sensing to restore the permuted data. Experiments on synthetic data, face images, educational and medical records reveal the potential of UPCA for applications such as data privatization and record linkage.

## 1 Introduction

### 1.1 Motivation

In principal component analysis, a cornerstone of machine learning and data science, one is given a data matrix $\tilde{X}$, assumed to be a corrupted version of a ground-truth data matrix $X^* = [x_1^* \cdots x_n^*] \in \mathbb{R}^{m \times n}$, typically but not necessarily assumed to have low rank, and the objective is to estimate $X^*$ or the column-space $S^* \subset \mathbb{R}^m$ of $X^*$. The most common types of corruptions that have attracted interest in modern studies are additive sparse perturbations [6, 47], outlier data points that lie away from $S^*$ [43, 40], and missing entries, the latter also known as low-rank matrix completion [7, 4].

Recently, permutations have been emerging as another type of data corruption, typically set in the context of linear regression, where the correspondences between the input and the output data have been partially distorted or even are entirely unavailable [38, 39, 22, 17, 30, 31, 46, 19, 41, 36]. There, one is given a point $x^*$ of a linear subspace $S^*$, but only up to a permutation of its coordinates, say $\tilde{x} = \Pi^* x^*$ with $\Pi^*$ an unknown permutation, and the goal is to find $x^*$ from the data $\tilde{x}, S^*$. This *Unlabeled Sensing* [38, 39] problem has many potential machine learning applications, e.g., record linkage [30, 31], visual [26, 27] or textual [5, 28, 29] permutation learning, and matching problems in neuroscience [21] and biology [1, 42].

While methods for unlabeled sensing rely on knowledge of the source subspace $S^*$, this is not always known in practice. On the other hand, data of the form $\tilde{X} = [\tilde{x}_1, \ldots, \tilde{x}_n] \in \mathbb{R}^{m \times n}$ with $\tilde{x}_j = \Pi_j^* x_j^*$ an unknown permutation of an unknown point $x_j^* \in S^*$, are often available, thus raising the question of whether $S^*$ can be estimated from $\tilde{X}$. An important example of this situation is record linkage [13, 20, 3], where the objective is to integrate data from independent sources, $\tilde{x}_1, \ldots, \tilde{x}_n \in \mathbb{R}^m$, for subsequent data analysis. Since the entries of different records $\tilde{x}_i$'s are collected separately, the data matrix $\tilde{X}$ is *unlabeled* in the sense that, the entries of its $i$-th row do not necessarily correspond

35th Conference on Neural Information Processing Systems (NeurIPS 2021).

to the same entity. Such kind of unlabeled data $\tilde{X}$ also arise in the context of data privatization, where the data provider anonymizes the original data $X^*$ by permuting each column of $X^*$ prior to release [11, 16]. Data re-identification is a concern since companies with privacy policies, health care providers, and financial institutions may release the collected data after anonymization. Understanding the fundamental limits of re-identifying the original data $X^*$ from the released ones $\tilde{X}$ is essential for striking a balance between data privacy and data preservation, the subject of a plenary talk in the 2019 International Conference on Machine Learning [2].

## 1.2 Contributions

In this paper we consider the recovery of $X^*$ from its unlabeled version $\tilde{X}$, which we term *Unlabeled Principal Component Analysis* (UPCA). We make the following contributions:

- We establish that as long as $r := \text{rank}(X^*) < \min\{m, n\}$ and $X^*$ is *generic* (see Definition 1), then up to a permutation of its rows, $X^*$ is the only matrix of rank less than or equal to $r$ that is compatible with $\tilde{X}$. This asserts that UPCA is a well-posed problem, since the inherent ambiguity of whether $\tilde{X}$ comes from $X^*$ or a row-permuted version of $X^*$ is in most cases practically harmless.

- We establish that in this basic formulation UPCA is a purely algebraic problem, by exhibiting a polynomial system of equations parametrized by $\tilde{X}$, whose solutions are all the row-permutations of $X^*$; solving the UPCA problem amounts to obtaining one such solution.

- Inasmuch as solving this polynomial system is in principle NP-hard, we introduce an efficient algorithmic pipeline for the practically relevant case where a significant part of the data have undergone the same permutation, while the rest of the points have been permuted arbitrarily; in the case of record linkage this would correspond to one of the records having much larger size than the others. The first stage of the pipeline employs PCA methods with robustness to outliers [43, 33, 25, 45, 37, 48, 18] to produce an estimate $\hat{S}$ of $S^*$ from $\tilde{X}$; the second stage of the pipeline uses unlabeled sensing methods [30, 32, 36, 23] to furnish an estimate $\hat{X}$ of $X^*$ from $\hat{S}$ and $\tilde{X}$.

- We introduce a simple but efficient algorithm for unlabeled sensing based on least-squares with recursive filtration (Algorithm 2).

- We assess our algorithmic pipeline on synthetic data, face images, educational and medical records, with encouraging results.

## 1.3 Related work

**Unlabeled sensing.** There is a large literature on application-specific problems that involve lack of correspondences, e.g. in computer vision or statistics; here we just review four recent methods for unlabeled sensing that will be used in this paper. In unlabeled sensing one is given a subspace $S^* \subset \mathbb{R}^m$ of dimension $r$ and a point $\tilde{x}$ which is some permuted version of a point $x^* \in S^*$ and the goal is to recover $x^*$ from $S^*$ and $\tilde{x}$. A critical distinction among methods in the literature is the sparsity level $\alpha$ of the permutation, that is the ratio of coordinates that are moved by the permutation.

The case of *dense* permutations ($\alpha = 1$) is extremely challenging, with existing methods only able to handle small ranks $r$. We consider two methods known to perform best in this regime. The first one is the algebraic-geometric AIEM method of [36]. This has linear complexity in $m$ and instead concentrates its effort on solving a polynomial system of $r$ equations in $r$ variables to produce an initialization for an expectation maximization algorithm. Currently, this method is efficient for $r \leq 5$ and intractable otherwise. A very different method is CCV-Min of [23], which proceeds via branch-&-bound together with concave minimization and can handle ranks $r \leq 8$.

For *sparse* permutations (small $\alpha$) dealing with higher ranks becomes possible [30, 32]. The $\ell_1$-RR algorithm of [30] applies an $\ell_1$ robust linear regression relaxation, and it works when $\alpha \leq 0.5$. Another approach is the Pseudo-Likelihood method (PL) of [32], which fits a two-component mixture density for each entry of $\tilde{x}$, one accounting for fixed data and the other for permuted data. The fitting is done via a combination of hypothesis testing, reweighed least-squares, and alternating minimization. Empirically, PL can tolerate up to $\alpha = 0.7$ but it is sensitive to the particular basis of $S^*$ used to generate $\tilde{x}$.

**Robust PCA with outliers.** PCA methods with robustness to outliers will also play a role in this paper. Among a large literature, we review four modern methods. In that context, it is assumed that $\tilde{X}$ can be partitioned into inlier points that lie in an unknown $r$-dimensional subspace $S^* \subset \mathbb{R}^m$ and outlier points that lie away from $S^*$, and the goal is to recover $S^*$ from $\tilde{X}$. The Outlier Pursuit (OP) method of [43] decomposes $\tilde{X}$ via convex optimization into the sum of low-rank and column-sparse parts. The convex method of [45], a successor of [33], referred to as Self-Expr, solves a self-expressive elastic net problem so that each $\tilde{x}_j$ is expressed as an $\ell_2$-regularized sparse linear combination of the other points. The self-expressive coefficients define transition probabilities of a random walk on a graph, and the average of the $t$-step transition probability distributions is used as a score for inliers vs. outliers, with higher scores expected for the former. The Coherence Pursuit (CoP) method of [25] is based on the following simple but effective principle: with $\tilde{X}_{-j}$ the matrix $\tilde{X}$ with column $j$ removed, for each $\tilde{x}_j$ one computes its coherence $\tilde{X}_{-j}^\top \tilde{x}_j$ with the rest of the points. As it turns out, inliers tend to have coherences of higher $\ell_2$-norm than outliers, and the top $r$ $\tilde{x}_j$'s are taken to span $\hat{S}$. Finally, the Dual Principal Component Pursuit (DPCP) of [37] solves an $\ell_1$ non-smooth problem on the Grassmannian via iterated-reweighed least-squares (IRLS), to compute an orthonormal basis for the orthogonal complement of the subspace. A dual formulation with a randomized singular value decomposition incorporated in the IRLS procedure is known as Fast Median Subspace [18]. In sharp contrast to other convex robust-PCA methods, DPCP has been shown in [48, 9] to tolerate as many outliers as the *square* of the number of inliers.

## 2 Theoretical Foundations

### 2.1 Problem Formulation

Let us denote by $\mathcal{P}_m$ the set of all permutations of coordinates of $\mathbb{R}^m$. We let $X^* = [x_1^* \cdots x_n^*] \in \mathbb{R}^{m \times n}$ be our ground-truth data matrix with rank $r < \min\{m, n\}$ and column-space $S^* = \mathcal{C}(X^*)$, and we suppose that the available data are

$$\tilde{X} = [\tilde{x}_1 \cdots \tilde{x}_n] = [\Pi_1^* x_1^* \cdots \Pi_n^* x_n^*] \in \mathbb{R}^{m \times n}, \tag{1}$$

where each $\Pi_j^* \in \mathcal{P}_m$ is an unknown permutation. Let $\mathcal{P}_m^n = \prod_{i \in [n]} \mathcal{P}_m$ be $n$ ordered copies of $\mathcal{P}_m$, where $[n] = \{1, \ldots, n\}$. For $\underline{\pi} = (\Pi_1, \ldots, \Pi_n) \in \mathcal{P}_m^n$ we set $\underline{\pi}(\tilde{X}) = [\Pi_1 \tilde{x}_1 \cdots \Pi_n \tilde{x}_n]$. We pose Unlabeled Principal Component Analysis (UPCA) as the following rank minimization problem:

$$\min_{\underline{\pi} \in \mathcal{P}_m^n} \ \text{rank} \, \underline{\pi}(\tilde{X}) \tag{2}$$

First, note that (2) never has a unique solution, because if $\underline{\pi} = (\Pi_1, \ldots, \Pi_n)$ is a solution, then so is $\underline{\pi}' = (\Pi\Pi_1, \ldots, \Pi\Pi_n)$, where $\Pi \in \mathcal{P}_m$ is any permutation. This reveals an inherent ambiguity of UPCA: it is only possible to recover $X^*$ from $\tilde{X}$ up to a permutation $\Pi X^*$ of its rows. On the other hand, this is rather harmless in many situations, since $\Pi X^*$ is the same dataset as $X^*$ except that the row-features appear now in some different order. Thus, our hope in formulating (2) is that the only solutions are of the form $\underline{\pi} = (\Pi\Pi_1^{*\top}, \ldots, \Pi\Pi_n^{*\top})$ with $\Pi$ ranging across $\mathcal{P}_m$ and $\Pi_j^*$ as in (1). However, without any other assumptions on the data $X^*$, there could in principle be additional undesired permutations that also give rank $X^*$, or even worse, the minimum rank in (2) could be lower than $r = \text{rank} \, X^*$. Our results show that for *generic* enough data, such pathological situations do not occur, and the only solutions to (2) are the ones associated with row-permutations of $X^*$.

### 2.2 Elements of Algebraic Geometry

Before stating our results, we make the notion of *generic* precise using some basic algebraic geometry [8, 15]. Let $Z = (z_{ij})$ be an $m \times n$ matrix of variables $z_{ij}$ and $\mathbb{R}[Z] = \mathbb{R}[z_{ij} : i \in [m], j \in [n]]$ the ring of polynomials in the $z_{ij}$'s with real coefficients. An *algebraic variety* of $\mathbb{R}^{m \times n}$ is the set of solutions of a polynomial system of equations in $\mathbb{R}[Z]$. In particular, the set of $(r + 1) \times (r + 1)$ determinants of $Z$ are polynomials in $z_{ij}$'s of degree $r + 1$ and define the algebraic variety

$$\mathcal{M}_r = \{X \in \mathbb{R}^{m \times n} | \text{rank} \, X \leq r\},$$

since a matrix $X \in \mathbb{R}^{m \times n}$ has rank at most $r$ if and only if all $(r + 1) \times (r + 1)$ determinants of $X$ are zero.

The algebraic variety $\mathcal{M}_r$ admits a topology, called *Zariski topology*, which makes it convenient to work with. The closed sets in this topology are the *algebraic subvarieties* of $\mathcal{M}_r$. These are sets of matrices of rank $\leq r$, which in addition satisfy certain other polynomial equations in $\mathbb{R}[Z]$. For example, the set of matrices of rank at most $r - 1$ is a proper closed subset of $\mathcal{M}_r$, because in addition to the equations defining $\mathcal{M}_r$, it is further defined by requiring all $r \times r$ determinants to be zero. *Open sets* in $\mathcal{M}_r$ are defined as complements of closed sets, or equivalently they are defined by requiring that certain sets of polynomials are not all simultaneously zero. For example, the set of matrices of rank exactly equal to $r$ is a proper open subset of $\mathcal{M}_r$ defined by the non-simultaneous vanishing of all $r \times r$ determinants of $Z$; a matrix has rank $r$ if and only if all $(r + 1) \times (r + 1)$ determinants are zero and least one $r \times r$ determinant is non-zero. Now, the algebraic variety $\mathcal{M}_r$ is *irreducible* in the sense that it can not be described as the union of two proper algebraic subvarieties of it. A consequence of this is that non-empty open sets of $\mathcal{M}_r$ have the very important property of being topologically dense. This means that given a non-empty open set $\mathcal{U} \subset \mathcal{M}_r$ and a point $X \in \mathcal{M}_r$, every neighborhood of $X$ intersects $\mathcal{U}$. It follows that under any non-degenerate continuous probability measure on $\mathcal{M}_r$, a non-empty Zariski-open set of $\mathcal{M}_r$ has measure $1$. For example, the set of matrices in $\mathcal{M}_r$ of rank $r$ is non-empty and open, and thus it is dense. Hence a randomly sampled matrix in $\mathcal{M}_r$ under a continuous probability measure will have rank $r$ with probability $1$. We refer to such a fact by saying that a generic matrix in $\mathcal{M}_r$ has rank $r$. More generally:

**Definition 1.** We say that a *generic* matrix in $\mathcal{M}_r$ satisfies a property, if the property is true for every matrix in a non-empty open subset of $\mathcal{M}_r$.

## 2.3 UPCA is a Well-Posed Problem

Our main theoretical result is[1]:

**Theorem 1.** *For $X^*$ a generic matrix in $\mathcal{M}_r$, we have that* $\operatorname{rank} \underline{\pi}(\tilde{X}) \geq r$ *for any* $\underline{\pi} \in \mathcal{P}_m^n$, *with equality if and only if* $\underline{\pi}(\tilde{X}) = \Pi X^*$ *for some* $\Pi \in \mathcal{P}_m$.

Theorem 1 says that for $X^* \in \mathcal{M}_r$ generic, and up to a permutation of the coordinates of $\mathbb{R}^m$, $S^*$ is the unique $r$-dimensional subspace that explains the data $\tilde{X}$ in the UPCA sense, and $r = \operatorname{rank} X^*$ is the minimum objective in (2).

## 2.4 UPCA is an Algebraic Problem

How can one go about solving the discrete optimization problem (2)? In general, brute force selection of the $\Pi_j$'s has complexity $\mathcal{O}\big((m!)^n\big)$, which is out of the question. On the other hand, problem (2) has a rich algebraic structure, which we harvest by showing that $X^*$, up to a permutation of its rows, arises as the unique solution to a polynomial system of equations.

To begin with, for each $j \in [n]$ and each $\ell \in [m]$, we define the following column-symmetric polynomials of $\mathbb{R}[Z]$:

$$\bar{p}_{\ell,j}(Z) := \sum_{i \in [m]} z_{ij}^\ell$$

$$p_{\ell,j}(Z) := \bar{p}_{\ell,j}(Z) - \bar{p}_{\ell,j}(\tilde{X})$$

Note that $\bar{p}_{\ell,j}\big(\underline{\pi}(Z)\big) = \bar{p}_{\ell,j}(Z)$ for any $\underline{\pi} \in \mathcal{P}_m^n$ and thus $\bar{p}_{\ell,j}(\tilde{X}) = \bar{p}_{\ell,j}(X^*)$. Now let us think of $X \in \mathcal{M}_r$ as a product of two matrices of size $m \times r$ and $r \times n$, and let us define another polynomial ring with variables associated to these two factors. For $i = r + 1, \ldots, m$, and $k \in [r]$ and $j \in [n]$, we let $b_{ik}, c_{kj}$ be a new set of variables over $\mathbb{R}$. Organize the $b_{ik}$'s to occupy the $(m - r) \times r$ bottom block of an $m \times r$ matrix $B$ whose top $r \times r$ block is the identity matrix of size $r$, and the $c_{kj}$'s into a $k \times n$ matrix $C = (c_{kj})$. For $i \in [m]$, we write $b_i^\top$ for the $i$-th row of $B$; for $j \in [n]$, we write $c_j$ for the $j$-th column of $C$. With $\tilde{x}_{ij}, x_{ij}^*$ the $i$-th coordinate of $\tilde{x}_j, x_j^*$ respectively, we obtain polynomials $q_{\ell,j}$ for $\ell \in [m]$, $j \in [n]$ of $\mathbb{R}[B, C]$ by substituting $z_{ij} \mapsto b_i^\top c_j$ in the $p_{\ell,j}(Z)$'s above:

$$q_{\ell,j}(B, C) := \bar{p}_{\ell,j}(BC) - \bar{p}_{\ell,j}(\tilde{X}) = \sum_{i \in [m]} (b_i^\top c_j)^\ell - \sum_{i \in [m]} \tilde{x}_{ij}^\ell = \sum_{i \in [m]} (b_i^\top c_j)^\ell - \sum_{i \in [m]} x_{ij}^{*\,\ell}$$

---

[1]The proofs of Theorems 1-3 can be found in [44].

The set of common roots of all $q_{\ell,j}$'s defines an algebraic variety that depends only on $X^*$, given by

$$\mathcal{Y}_{X^*} := \big\{(B', C') \in \mathbb{R}^{m \times r} \times \mathbb{R}^{r \times n} \,|\, q_{\ell,j}(B', C') = 0, \,\forall \ell \in [m], \,\forall j \in [n]; \;\; B'_{[r],[r]} = I_r\big\},$$

where $B'_{[r],[r]} = I_r$ signifies that the top $r \times r$ block of $B' \in \mathbb{R}^{m \times r}$ is the identity matrix. Let us get a feeling about the points of $\mathcal{Y}_{X^*}$. With $\Pi \in \mathcal{P}_m$, if the column-space $\mathcal{C}(\Pi X^*)$ of $\Pi X^*$ does not drop dimension upon projection onto the first $r$ coordinates, then there exists a unique basis $B_\Pi^*$ of $\mathcal{C}(\Pi X^*)$ with the identity matrix occurring at the top $r \times r$ block. In that case, there is a unique factorization $\Pi X^* = B_\Pi^* C_\Pi^*$ and the point $(B_\Pi^*, C_\Pi^*)$ lies in the variety $\mathcal{Y}_{X^*}$ because

$$q_{\ell,j}(B_\Pi^*, C_\Pi^*) = \bar{p}_{\ell,j}(B_\Pi^* C_\Pi^*) - \bar{p}_{\ell,j}(\tilde{X}) = \bar{p}_{\ell,j}(\Pi X^*) - \bar{p}_{\ell,j}(X^*) = \bar{p}_{\ell,j}(X^*) - \bar{p}_{\ell,j}(X^*) = 0.$$

Our second result says that if $X^*$ is generic, then all points of $\mathcal{Y}_{X^*}$ are of this type. That is, they correspond to factorizations $B_\Pi^* C_\Pi^*$ of $\Pi X^*$ as $\Pi$ varies across all permutations:

**Theorem 2.** *For a generic matrix $X^*$ in $\mathcal{M}_r$ we have*

$$\mathcal{Y}_{X^*} = \big\{(B_\Pi^*, C_\Pi^*) \in \mathbb{R}^{m \times r} \times \mathbb{R}^{r \times n} \,|\, \Pi \in \mathcal{P}_m; \; B_{\Pi,[r],[r]}^* = I_r; \; \Pi X^* = B_\Pi^* C_\Pi^*\big\}.$$

Thanks to Theorem 2 we have the following important conceptual finding. Assuming $X^*$ is generic, to obtain $X^*$ up to some permutation of its rows from $\tilde{X}$, one needs to compute an arbitrary root $(B', C')$ of the polynomial system of equations

$$q_{\ell,j}(B, C) = 0, \,\forall \ell \in [m], \,\forall j \in [n] \tag{3}$$

and multiply its factors to get $B'C'$. Developing a polynomial system solver for UPCA would involve two main challenges: attaining robustness to noise and scalability, with the former typically easier to deal with than the latter. We leave such an endeavor to future research.

## 2.5 UPCA with Dominant Permutations

We close this section with a special case of interest, where part of the data have undergone the same *dominant* permutation; in fact, given the inherent ambiguity of UPCA discussed above, we may as well take this dominant permutation to be the identity matrix $I_m$ of size $m \times m$. To make this precise, we define the multiplicity $\mu(\Pi)$ of a permutation $\Pi \in \mathcal{P}_m$ to be the number of times that $\Pi$ appears as $\Pi = \Pi_j^*$ in (1) with $j$ ranging in $[n]$. We have:

**Theorem 3.** *Suppose that $\mu(I_m) \geq r + 1$ while $\mu(\Pi) < r$ for any other $\Pi \neq I_m$. Then for a generic $X^* \in \mathcal{M}_r$, we have that $S^*$ is the unique solution to the following consensus maximization problem*

$$\max_{\dim S \leq r} \#\{\tilde{x}_j \,|\, \tilde{x}_j \in S; j \in [n]\}, \tag{4}$$

*where $\#$ denotes the cardinality of a set, and the maximization is taken over all subspaces $S \subset \mathbb{R}^m$ of dimension $\leq r$.*

Theorem 3 says that for sufficiently generic ground truth data $X^*$, the given data $\tilde{X}$ admit a natural partition into a set of inliers and outliers with respect to the linear subspace $S^*$:

$$\tilde{X}_{\text{in}} := \{\tilde{x}_j \,|\, \tilde{x}_j \in S^*\}, \quad \tilde{X}_{\text{out}} := \{\tilde{x}_j \,|\, \tilde{x}_j \notin S^*\}$$

Of course we do not know what the partition into inliers and outliers is, because we do not know what $S^*$ is. But the presence of this geometric structure is enough for PCA methods with robustness to outliers to operate on $\tilde{X}$ in order to estimate $S^*$. The rest of the paper proceeds algorithmically building on this insight.

## 3 Algorithms

### 3.1 Two-stage algorithmic pipeline for UPCA

We saw in the previous section that the UPCA problem (2) is well-defined (Theorem 1) and in principle solvable by a polynomial system of equations (Theorem 2). However, this polynomial system is at the moment intractable to solve even for moderate dimensions. On the other hand, Theorem 3 suggests the following algorithmic pipeline, for the case where there is a dominant

permutation, which we can take to be the identity as in the theorem. At Stage-I of the pipeline, a PCA method with robustness to outliers [43, 33, 25, 45, 37, 48, 18] is employed to produce an estimate $\hat{S}$ of $S^*$ from $\tilde{X}$. At Stage-II of the pipeline, one feeds $\hat{S}$ and $\tilde{X}$ to an unlabeled sensing method [30, 32, 36, 23] which operates point by point, returning for every $\tilde{x}_j$ an estimate $\hat{x}_j$ of $x_j^*$. Here one may choose to threshold the $\tilde{x}_j$'s based on their distance to $\hat{S}$ and apply unlabeled sensing on the outliers only. Alternatively, if extra computational power is available for dispensing with choosing a threshold, one may apply unlabeled sensing on every $\tilde{x}_j$; it is this approach that we follow in the rest of the paper. This procedure is summarized in Algorithm 1.

---

**Algorithm 1** Two-stage Algorithmic Pipeline for UPCA

---

1: **Input:** observed data matrix $\tilde{X}$, rank $r$
2: estimate $\hat{S}$ of $S^* \leftarrow$ outlier-robust PCA on $\tilde{X}$               ▷ Stage-I
3: **for** $j = 1, \ldots, n$ **do**                                 ▷ Stage-II
4:      estimate $\hat{x}_j$ of $x_j^* \leftarrow$ unlabeled sensing on $(\tilde{x}_j, \hat{S})$
5: **end for**
6: **return** $\hat{X} = [\hat{x}_1, \ldots, \hat{x}_n]$ estimate of $X^*$

---

### 3.2 A new method for unlabeled sensing: Least-Squares with Recursive Filtration (LSRF)

Inasmuch as there are very few scalable unlabeled sensing methods, we here propose a simple but comparatively efficient alternative, Algorithm 2. This method is parameter-free. It alternates between ordinary least-squares and a dimensionality reduction step that removes the coordinate of the ambient space on which the residual error attains its maximal value, until $r$ coordinates are left. The number of iterations is fixed as $(m - r)$, so the overall complexity of Algorithm 2 is $\mathcal{O}(m^2 r^2)$. We here also mention the time complexity of other unlabeled sensing methods for comparison: AIEM has complexity approximately $\mathcal{O}(m + r^r)$, PL has complexity at least $\mathcal{O}(kmr^3)$ via second-order optimization, and $\ell_1$-RR has complexity at least $\mathcal{O}(mr^3 + k(m + r^2))$ via sub-gradient descent, where $k$ is the number of iterations.

---

**Algorithm 2** Unlabeled Sensing via Least-Squares with Recursive Filtration (LSRF)

---

1: **Input:** permuted point $\tilde{x}_j$, basis $B^*$ of subspace $S^*$
2: $v^{(0)} \leftarrow \tilde{x}_j, A^{(0)} \leftarrow B^*$
3: **for** $k = 1, \ldots, m - r$ **do**
4:      $c \leftarrow A^{(k-1)^\dagger} v^{(k-1)}$
5:      $i' \leftarrow \text{argmax}_i |v_i^{(k-1)} - A_i^{(k-1)} c|$
6:      remove the $i'$-th entry of $v^{(k-1)}$ to get $v^{(k)}$
7:      remove the $i'$-th row of $A^{(k-1)}$ to get $A^{(k)}$
8: **end for**
9: **return** $\hat{x}_j = A^{(m-r)} A^{(m-r)^\dagger} v^{(m-r)}$ estimate for $x_j^*$

---

## 4 Experimental Evaluation

### 4.1 Robust-PCA with permutation-induced outliers

We begin by assessing the performance of Stage-I of the pipeline. This entails understanding how different PCA methods with robustness to outliers behave when the outliers are induced by permutations, as in Theorem 3. We consider Self-Expr [45, 33], CoP [25], OP [43], and DPCP [37]; see related work in section 1. We fix $m = 50$, $n = 500$. With $\dim S^* = r = 1 : 1 : 49$, we sample $S^*$ at random from the Grassmannian $\text{Gr}(r, m)$. Then $n$ points $x_j^*$ are sampled at random from the intersection of $S^*$ with the unit sphere of $\mathbb{R}^m$ to yield $X^*$. Let $n_{\text{in}}$ be the number of inliers $\tilde{X}_{\text{in}}$ and $n_{\text{out}}$ the number of outliers $\tilde{X}_{\text{out}}$, with $n_{\text{in}} + n_{\text{out}} = n$. We consider outlier ratios $n_{\text{out}}/n = 0.1 : 0.1 : 0.9$. With fixed $n_{\text{out}}/n$, we set $\Pi_j^*$ to the identity for $j \in [n_{\text{in}}]$ and set $\Pi_j^*$'s for $j > n_{\text{in}}$ as follows. We consider both dense ($\alpha = 1$) and sparse ($\alpha = 0.1$) permutations. For a fixed $\alpha$, we obtain $\Pi_j^*$

by randomly choosing $\alpha m$ coordinates for each $x_j^*$ and applying a random permutation on those coordinates. As evaluation metric we use the largest among all $r$ principal angles between $\hat{S}$ and $S^*$, denoted by $\theta_{\max}(S^*, \hat{S})$. Recall $\theta_{\max}(S^*, \hat{S}) = 0$ if and only if $\hat{S} = S^*$.

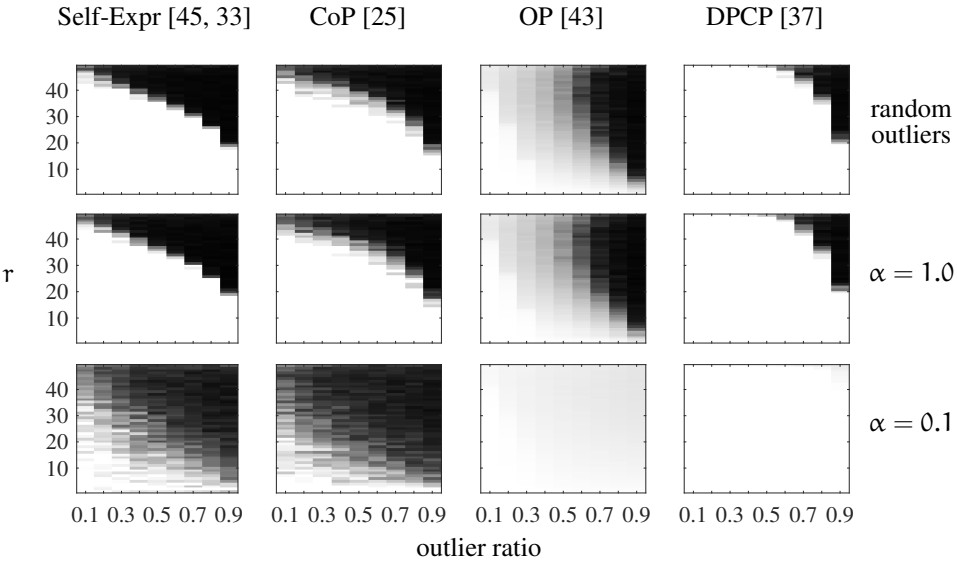

Figure 1: $\theta_{\max}(S^*, \hat{S})$ in Stage-I: outlier ratio vs. rank phase transitions for various PCA methods with robustness to outliers.

Figure 1 depicts the *outlier-ratio versus rank* phase transitions, where to calibrate the analysis with what we know about these methods from prior work, we have included in the top row of the figure the phase transitions for outliers randomly chosen from the unit sphere. By reading that top row we recall: i) DPCP has overall the best performance across all ranks and all outlier ratios, ii) OP identifies correctly $S^*$ only in the low rank low outlier-ratio regime, as expected from its conceptual formulation, and iii) CoP and Self-Expr, even though low-rank methods in spirit, they have accuracy similar to each other and considerably better than OP. We also note that CoP is the fastest method requiring $0.51sec$ for the computation of a single phase transition plot, Self-Expr is the slowest with $752sec$, and DPCP and OP take $1.31sec$ and $5.62sec$, respectively[2].

Now let us look at what happens for permutation-induced outliers. For $\alpha = 1$, where the permutations move all the coordinates of the points they are corrupting, we see that the phase transition plots are practically the same as for random outliers. In other words, obtaining the outliers by randomly permuting all coordinates of inlier points, with different permutations for different outliers, seems, even for low subspace dimensions, to be yielding an outlier set as generic for the task of subspace learning as sampling the outliers randomly from the unit sphere. A second interesting phenomenon is observed when the permutation ratio is decreased to $\alpha = 0.1$. In that regime the methods exhibit two very different trends. On one hand, CoP and Self-Expr appear to break down, which is expected, because as the permutations become more sparse, the outlier points become more coherent with the rest of the data set. On the other hand, the accuracy of DPCP and OP improves for sparser permutations; a justification for this is that both methods get initialized via the SVD of $\tilde{X}$, which yields a subspace closer to $S^*$ for smaller $\alpha$. An interesting research direction is to analyze the theoretical guarantees of these methods for this specific type of outliers.

## 4.2 UPCA on synthetic data

Next, we evaluate the UPCA pipeline of Algorithm 1 on synthetic data. We keep $m = 50$ as before, and add spherical noise corresponding to a fixed SNR of 40dB. We get the estimate $\hat{S}$ of $S^*$ via DPCP [37] in Stage-I and apply the unlabeled sensing methods [36, 23, 30, 32] and Algorithm 2 in Stage-II to get $\hat{X}$ from $\hat{S}$ and $\tilde{X}$. We distinguish between dense and sparse permutations.

---

[2]Experiments are run on an Intel(R) i7-8700K, 3.7 GHz, 16GB machine.

**UPCA with dense permutations ($\alpha = 1$).** Figure 2 depicts the relative estimation error of $\hat{X}$ for different outlier ratios from 75% (25 inliers) to 94% (6 inliers) and ranks $r = 3, 4, 5$. To assess the overall effect of the quality of $\hat{S}$, we use two versions of AIEM and CCV-Min. The first, denoted by AIEM($\hat{S}$) and CCV-Min($\hat{S}$), uses as input the estimated subspace $\hat{S}$, while the second version, AIEM($S^*$) and CCV-Min($S^*$), uses the ground-truth subspace $S^*$. Note that the estimation error of AIEM($S^*$)/CCV-Min($S^*$) is independent of the outlier ratio. On the other hand, the estimation error of AIEM($\hat{S}$)/CCV-Min($\hat{S}$) depends on the outlier ratio through the computation of $\hat{S}$. Indeed, $\hat{S}$ is expected to be closer to $S^*$ for smaller outlier ratios, as we already know from Figure 1. In particular, for up to 75% outliers the estimation error of AIEM($\hat{S}$)/ CCV-Min($\hat{S}$) coincides with that of AIEM($S^*$)/ CCV-Min($S^*$), indicating an accurate estimation of $S^*$. At the other extreme, for 94% outliers both AIEM($\hat{S}$)/ CCV-Min($\hat{S}$) break down, indicating that the estimation of $\hat{S}$ failed. Finally, note that CCV-Min has at least half order of magnitude smaller estimation error than AIEM. This is due to our specific choice of the branch-&-bound CCV-Min parameters which control the trade-off between accuracy and running time; for example, for $r = 3$ and 75% outliers, AIEM runs in $0.042sec$ with 1% error, while CCV-Min needs about $15sec$ to bound $\hat{X}$ 0.42% away from $X^*$.

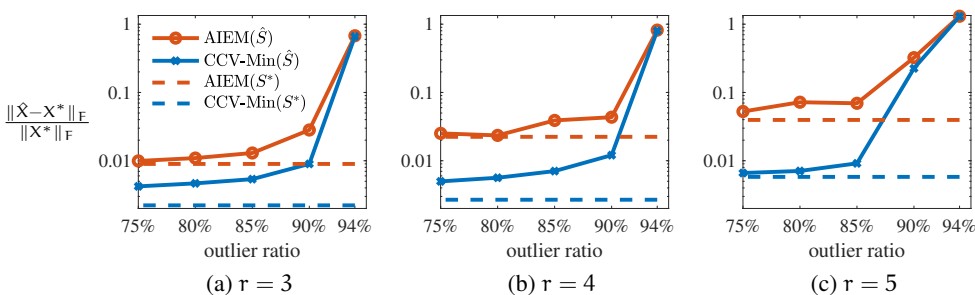

Figure 2: UPCA (Algorithm 1) for dense permutations ($\alpha = 1$) with $\hat{S}$ produced by DPCP [37] at Stage-I and $\hat{X}$ produced by AIEM [36] or CCV-Min [23] at Stage-II.

**UPCA with sparse permutations ($\alpha \leq 0.6$).** Figures 3b-3d show the relative estimation error of UPCA for $\alpha = 0.1 : 0.1 : 0.6$, rank $r = 1 : 1 : 25$, and outlier ratio fixed to 90%, with $\hat{S}$ computed in Stage-I by DPCP [37] and $\hat{X}$ computed via $\ell_1$-RR [30], PL [32] or Algorithm 2 from $\tilde{X}$ and $\hat{S}$ in Stage-II. It is important to note that $r = 25 = m/2$ is the largest rank for which unique recovery of $X^*$ is theoretically possible [38, 39, 34, 35, 10, 24]. Figure 3a shows that $\theta_{max}(S^*, \hat{S})$ always stays below $2°$, indicating the success of DPCP. Now $\ell_1$-RR and Algorithm 2 have similar accuracy, but Algorithm 2 is more efficient than $\ell_1$-RR, considering that computing $\hat{X}$ takes $0.3sec$ for Algorithm 2 and $1.5min$ for $\ell_1$-RR. Even though PL delivers $\hat{X}$ in $1sec$, it is not performing as well, which we attribute to its sensitivity on the particular basis of $S^*$ that is used to generate the data; this is not available here since DPCP returns the specific basis of dual principal components.

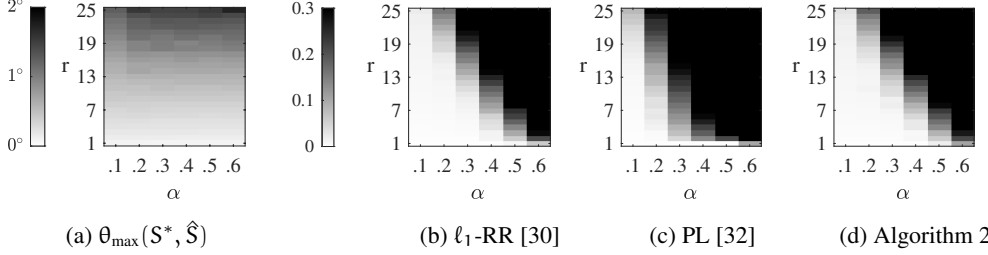

Figure 3: Estimation error $\frac{\|X^* - \hat{X}\|_F}{\|X^*\|_F}$ of UPCA (Algorithm 1) for sparse permutations ($\alpha \leq 0.6$) and outlier ratio 90%, with $\hat{S}$ computed by DPCP [37] in Stage-I and $\hat{X}$ computed by $\ell_1$-RR [30], PL [32] or Algorithm 2 in Stage-II.

## 4.3 UPCA on face images

In this section we offer a flavor of how the ideas discussed so far apply in a high-dimensional example with real data. We use the well-known database Extended Yale B [14], which contains fixed-pose face images of distinct individuals, with 64 images per individual under different illumination conditions. It is well-established that the images of each individual approximately span a low-dimensional subspace. It turns out that for our purpose the value $r = \dim S^* = 4$ is good enough. Since each image has size $192 \times 168$, the images of each individual can be approximately seen as $n = 64$ points $x_j^*, j \in [64]$ of a 4-dimensional linear subspace $S^*$, embedded in an ambient space of dimension $m = 32256$. In what follows we only deal with the images of a fixed individual. We consider four permutation types corresponding to fully or partially ($\alpha = 0.4$) permuting image patches of size $16 \times 24$ or $48 \times 42$, as shown in the second column of Figure 4. To generate a fixed number of $n_{out} = 16$ outliers, only one out of the four permutation types is used for each trial. The original images (inliers) together with the ones that have undergone patch-permutation (outliers) are given without any inlier/outlier labels, and the task is to restore all corrupted images. This is a special case of visual permutation learning, recently considered using deep networks [26, 27].

original    outlier    AIEM[36]    $\ell_1$-RR[30]    PL[32]  Algorithm 2

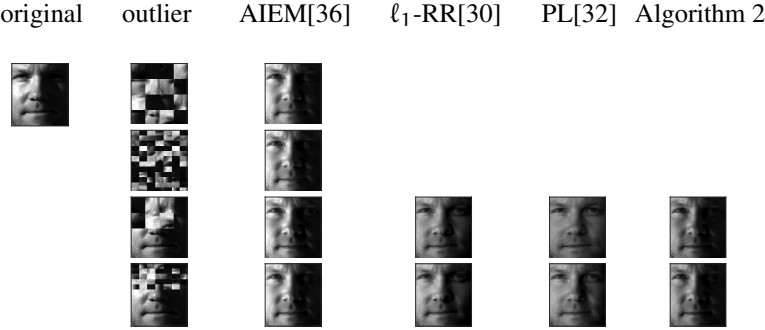

Figure 4: UPCA on the face dataset Extended Yale B.

The first column of Figure 4 shows an original image, and the second column shows the corresponding outlier obtained by applying a sample permutation for each of the four different permutation types. Columns three to six give the corresponding point in the output of Algorithm 1 for different unlabeled sensing methods and $\hat{S}$ computed by DPCP [37] (CCV-Min [23] is not included as branch-&-bound becomes prohibitively expensive for such large $m$). Notably, AIEM [36] rather satisfactorily restores the original image regardless of permutation type. The performance of the other three methods is shown only for their operational regime of sparse permutations, and Algorithm 2 most accurately captures the illumination of the original image. Overall, we find these results encouraging, especially if one takes into consideration that the methods are very efficient, requiring only $0.2sec$ (AIEM), $7sec$ ($\ell_1$-RR), $0.2sec$ (PL) and $10sec$ (Algorithm 2), discounting the DPCP step, which costs $0.1sec$, regardless of permutation type. This is in contrast with existing deep network architectures for visual permutation learning, such as [27], which are based on branch-&-bound and thus have in principle an exponential complexity in the number of permuted patches.

## 4.4 UPCA on data re-identification

Finally, we evaluate the UPCA Algorithm 1 for the task of data re-identification (see section 1) using real educational and medical records and simulated permutations for various sparsity levels $\alpha$, thus emulating a privacy protection scenario. Both of the datasets that we use contain no personally identifiable information. DPCP [37] computes $\hat{S}$ in Stage-I, and $\ell_1$-RR [30], PL [32], or Algorithm 2 produces $\hat{X}$ in Stage-II.

The first dataset consists of the test scores of $m = 707$ high-school students on 6 subjects during two different periods, together with the sum of the score tests for each period, thus $n = 14$. For 7 out of 14 tests, we apply random permutations of the student indices and thus have 50% outliers. With $r = 3$, the relative estimation errors on the score records are shown in Figure 5a. The black dashed line depicts the relative difference between the observed data $\tilde{X}$ and the original data $X^*$, which as expected increases for higher $\alpha$'s. Since $\ell_1$-RR, PL, and Algorithm 2 are by design suitable for sparse

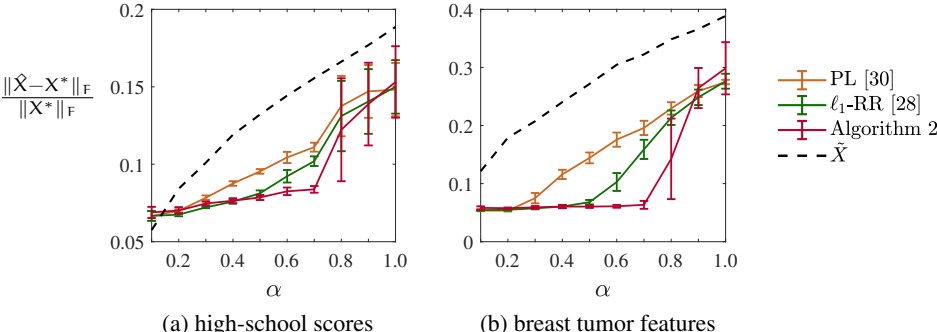

$$\frac{\|\hat{X}-X^*\|_F}{\|X^*\|_F}$$

(a) high-school scores      (b) breast tumor features

Figure 5: Relative estimation error on real data in de-anonymization.

permutations, their performance naturally degrades for large $\alpha$. But up to $\alpha = 0.7$ it aligns with our earlier findings in that Algorithm 2 tends to have superior performance on the average, and we also see that in that regime it also has the smallest variance.

The second dataset consists of all the benign cases in Breast Cancer Wisconsin (Diagnostic) [12]. It has $m = 357$ patients and $n = 30$ features of a breast mass digitized image for each patient. We randomly permute the patient indices for 15 of the features thus having 50% outliers and set $r = 4$. Figure 5b shows the relative estimation error of $\hat{X}$ for various permutation sparsity levels $\alpha$, with the unlabeled sensing methods exhibiting the same trend as before. Remarkably, for $\alpha = 0.7$, the UPCA Algorithm 1 incorporating Algorithm 2 in Stage-II reduces the original error of the data $\tilde{X}$ from 32.24% to 6.35% in $0.5sec$, as opposed to 15.90% and 19.57% when $\ell_1$-RR [30] or PL [32] are incorporated, respectively.

## 5 Discussion

Some interesting conclusions can be drawn from this work regarding privacy protection. First, it appears to be not secure for data providers to only partially permute the data, since, as we have seen, permuting only a subset of the features enables re-identification. Re-identification is also possible if a database is fully permuted but another uncorrupted database with the same subjects is accessible or present in an assembled dataset. On the other hand, arbitrary dense permutations are extremely difficult to recover, as one would need to solve an intractable polynomial system.

A related interesting direction for future research is the case where on top of permutations one has missing entries, a problem that we term unlabeled matrix completion. The missing entries, may either arise naturally, since for example not all same subjects are common across databases, but they could also serve as part of a privacy protection protocol. [44] discusses the algebraic structure of unlabeled matrix completion and gives results regarding finite recovery.

## Acknowledgement

This work was funded by ShanghaiTech start-up grant 2017F0203-000-16.

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
