# OpenReview forum: "Unlabeled Principal Component Analysis"
_NeurIPS.cc/2021/Conference — NeurIPS 2021 Poster_

### Official Review · Reviewer_9vKy · 2021-07-15

**Rating:** 6
**Confidence:** 3

**Summary:**

This paper introduces unlabeled principal component analysis (UPCA), which refers to the reconstruction of the ground-truth data matrix X^* from its unlabeled version \tilde{X}, whose each column is generated from a permutation of the coordinates of the corresponding column of X^*. The authors first establish that UPCA is well-posed by showing that if X^* is sufficiently "generic", then the solution to the UPCA presented in Eq. (2) applied to \tilde{X} is X^* (up to a global permutation matrix). The authors then establish that X^* (up to a global permutation matrix) is the unique solution of a polynomial system of equations parametrized by \tilde{X}. Based on these theoretical results, the authors propose a two-stage algorithmic pipeline for UPCA for a practically relevant case where only a fraction of the data has been permuted. The first stage uses robust PCA methods to estimate \hat{S} of the ground-truth column space S^* from \tilde{X}. The second stage uses unlabeled sensing methods to estimate \hat{X} of the ground-truth data matrix X^* from \hat{S} and \tilde{X}. The authors also introduce a novel algorithm for unlabeled sensing based on least-squares with recursive filtration (LSRF).

**Limitations And Societal Impact:**

Yes, the limitations and potential negative societal impact of this work have been addressed in the Checklist.

**Main Review:**

Overall, I think the introduced UPCA problem is interesting with both theoretical guarantees provided in Theorems 1 to 3 and potential practical applications as demonstrated in Section 4, and the technical results seem to be correct (but I have not checked the proofs in detail). Some comments:

1) The notion "generic" is important for the theorems and the authors should present a Definition 1 to provide a clear and concise definition of it. They should refer to Definition 1 when "generic" first appears in Line 45 on Page 2.

2) The authors should present the Output of Algorithms 1 and 2 clearly.

3) In Figures 3 and 4, Algorithm 2 can be abbreviated as LSRF, and the authors do not need to present the reference for each algorithm in all the figures.

4) Currently, from the experimental section, I cannot see the benefit of introducing Algorithm 2 (LSRF) clearly. I guess the major advantage of LSRF is its efficiency. But the authors only mention that it is fast for a single case corresponding to Figure 3. For a case corresponding to Figure 4, it is even the slowest algorithm (takes 10 seconds, as mentioned in Line 288). To show LSRF is more efficient than other unlabeled sensing methods to compare, the authors should present the time complexity of all these methods clearly (and perhaps the parameter regimes where these methods work) and present tables comparing the running time of these methods for many cases.

5) Some notational issues: Line 34, revise \tilde{x}_1 \in \mathbb{R}^m, \ldots, \tilde{x}_n  \in \mathbb{R}^m to \tilde{x}_1, \ldots, \tilde{x}_n  \in \mathbb{R}^m; Line 35, revise ith row to $i$-th row. Line 98, it is weird to write S^* = C(X^*) without any definition of C. Line 250, avoid using msec. Lines 258 and 289, avoid using 0.3sec seconds and 10sec seconds.




**Time Spent Reviewing:**

7

---

> ### Author Response · Authors · 2021-08-08
> **Response to Reviewer 9vKy**
>
> We thank the reviewer for finding the problem interesting, acknowledging the theoretical guarantees and the potential for practical applications. We also thank the reviewer for suggesting notational improvements; we will follow them. We will also add the definition of “generic” and the outputs of the algorithms. Finally, we thank the reviewer for suggesting a complexity and parameter analysis. If the paper is accepted, we will have an additional page in which we will include a table containing the complexities of the algorithms. Let us mention them here: i) AIEM is exponential in $r$ and linear in $m$, 3) Algorithm 2 has complexity $O(m^2 r^2)$, iii) each iteration of PL has complexity at least $O(m r^3)$ if done via second-order optimization and iv) $\ell_1$-RR has complexity at least $O(m r^2 + k(m+r^2))$ if done via sub-gradient descent, where $k$ is the number of iterations.
>
> Regarding the performance of Algorithm 2: please see our response to reviewer Xvq6, part 2. A parameter analysis is also very important, and we will add it in a longer version of this paper.

---

> > ### Comment · Reviewer_9vKy · 2021-09-01
> > **Responses to the authors**
> >
> > Thanks for the authors' responses. I still think the theoretical guarantees in this paper are interesting. But currently, although the proofs in the supplementary material are not long, it is still quite tiring for me to remember the notation and follow the logic/connections between steps and auxiliary results. I would like to keep my score unchanged and I hope the authors can provide proof sketches or intuitions in their revised version.

---

### Official Review · Reviewer_8z2R · 2021-07-16

**Rating:** 7
**Confidence:** 4

**Summary:**

The paper aims to recover a low-rank matrix in which the entries of the columns are permuted. The paper first provided theoretical analysis to show the feasibility of the recovery problem and then proposed a two-step method with a modified unlabeled sensing algorithm. The numerical results on synthetic data, face images, and other two datasets showed that the proposed two-step method is quite effective and the modified unlabeled sensing algorithm can outperform existing algorithms in most cases.

**Limitations And Societal Impact:**

Yes.

**Main Review:**

In the proposed method, the first step is to perform robust PCA on the permuted data matrix to find an estimation of the subspace.  The second step is to use the estimated subspace basis to perform unlabeled sensing to recover the matrix.  The author investigated the performance of four RPCA algorithms and four unlabeled sensing algorithms including the one proposed in the paper. In general, the paper consists of theory, practical algorithm, and real applications. The paper is well-written and the idea is quite interesting and novel.  Nevertheless, there are a few limitations.

1. It seems that there is a big "gap" between Section 2 and Section 3. The proposed algorithm does not consider or take advantage of any information in Theorem 2/3. Therefore, the significance of Theorem 2/3 is not big.

2. It is surprising that in Figure 1 DPCP works well when the rank is 48 or 49. In these cases, the matrix X is almost full-rank.

3. In Figure 4, why did Algorithm 2 fail in the first two cases? The performance is quite different from that in Section 4.4.

**Time Spent Reviewing:**

3

---

> ### Author Response · Authors · 2021-08-08
> **Response to Reviewer 8z2R**
>
> We thank the reviewer for acknowledging the strength of our paper. The reviewer mentioned three apparent limitations, but as we now explain, these are not real limitations.
>
> 1. “The proposed algorithm does not consider or take advantage of any information in Theorem 2/3. Therefore, the significance of Theorem 2/3 is not big.” On the contrary, the proposed algorithmic pipeline (Algorithm 1) is directly inspired by Theorem 3, which asserts that there is a unique subspace of dimension r that agrees with the data, and thus it is meaningful to search for this subspace. Besides, Theorems 1 and 2 are both of fundamental significance in understanding the problem of UPCA. Since UPCA is a problem that has not been studied before, we believe these theorems are important.
> 2. “It is surprising that in Figure 1 DPCP works well when the rank is 48 or 49. In these cases, the matrix X is almost full-rank.” DPCP is a robust-PCA method specifically designed for subspaces of high relative dimension $\frac{r}{m}$ and in particular for hyperplanes. It has already been established in [34] and in subsequent work that DPCP can separate inliers and outliers successfully even when the dimensionality of the inliers is one less than the ambient dimension (the case of hyperplanes); this is row 1 of Fig. 4 where the outliers are random points. Row 2 of Fig. 4 shows that DPCP has a similar ability for outliers coming from dense permutations and row 3 shows that the performance is even better when the permutations are sparse. The text gives intuition as to why this is the case.
> 3. “In Figure 4, why did Algorithm 2 fail in the first two cases? The performance is quite different from that in Section 4.4.” Please see our response to reviewer Xvq6, part 2.

---

### Official Review · Reviewer_Xvq6 · 2021-07-23

**Rating:** 7
**Confidence:** 4

**Summary:**

- The authors describe and tackle the problem of unlabelled principal component analysis, a variant of PCA where the columns of the data matrix might have been permuted arbitrarily with different permutations for each column. The problem shares some of its motivation with the easier problem of unlabelled sensing, where the principal subspace is known.
- The authors divide the problem of UPCA into two natural steps: i) Identifying a rank-r subspace S* where all data points lie if permitted correctly, and; ii) mappings the permuted data points onto the identified subspace. These two steps can be naturally addressed by existing lines of work: robust PCA for former and unlabelled sensing for the latter.
- The authors use a algebraic framework to formulate and solve the problem. The main theoretical result is a powerful one: under genericity of the true solution, then any solution of the rank minimization formulation of the problem recovers the ground truth up to a permutation.
- A combinatorial solution is prohibitively expensive. One could consider a polynomial system solver, but the authors correctly point out that scalability and robustness to noise would pose separate challenges and leave that to future work.
- Instead, the authors focus on specific subclasses of the UPCA problem where only a fraction of the entries are permuted. Under this assumption it is indeed possible to have efficient solutions of the problem, and the proposed two-stage workflow seems well suited.
- Finally, the authors provide extensive experiments on their methodology on synthetic and real datasets.

**Limitations And Societal Impact:**

The authors mention some positive aspects when it comes to societal impact, but I feel like there’s an important point missing:
- Is it not true that powerful de-anonymization methodology can be used to violate the privacy of certain data releases? That sounds potentially harmful, so it might be useful to give it some thought and add a bit of discussion.

**Main Review:**

I recommend acceptance for the paper. In my opinion it is clearly above the bar in all respects. The problem is convincingly useful, challenging and not really studied so far. The authors put a lot of effort providing not just the raw technical results but also very helpful pedagogical discussion that makes this mathy paper more easily accessible to the ML folk.

Strong points:
1. Motivation: This line of work and this problem in particular is relatively new and not necessarily widely recognized as important. So it was important for the authors to provide convincing motivation. I was happy to see that the introduction provides a solid, clear motivation for the problem with plenty of references.
2. Technically the paper is solid. The authors use unusually heavy algebra to prove their main result, but Section 2 provides a concise, but pedagogical explanation of the algebraic results necessary in the paper.
3. The evaluation is relatively extensive and more than sufficiently thorough. I appreciated the effort that the authors put in evaluating different base methods for the two “stages” for their UPCA workflow.

Weak points and questions:
1. With the exception of the proposal of Algorithm 2 (an alternative method for unlabelled sensing) the methodology in the paper is completely drawn from the robust PCA and unlabelled sensing literature. I mention this point for completeness, but I think that it is a minor point that does not detract from the paper’s value. The strong points mentioned above outweigh this point.
2. Algorithm 2 seems to perform great on the re-identification problem of Section 4.4. On the other hand, it doesn’t seem particularly useful in the face experiments of 4.3 (AIEM appears to be both faster and more accurate that Algorithm 2  in the experiment of Figure 4). Can the authors comment a bit on what might be the best conditions to use Algorithm 2 vs AIEM?
3. There is discussion in the main body of the paper on what it means for a solution to be generic in algebraic terms. Can the authors provide some discussion on a few reasonable necessary or sufficient conditions for data matrices to be generic in applications?

**Time Spent Reviewing:**

5

---

> ### Author Response · Authors · 2021-08-08
> **Response to Reviewer Xvq6**
>
> We thank the reviewer for acknowledging the strength of this paper. We are also pleased the reviewer appreciated the effort we put in the writing. We address the reviewer’s concerns:
>
> 1. “the methodology in the paper is completely drawn from the robust PCA and unlabelled sensing literature”. This is true regarding the algorithmic methodology, as we clearly explain in the paper, so the reviewer is absolutely right in this regard. Regarding the theoretical part and the proofs though, we are not aware of many papers in robust-PCA and unlabeled sensing literature with a similar methodology.
>
> 2. “Can the authors comment a bit on what might be the best conditions to use Algorithm2 vs AIEM?” In unlabeled sensing, the complexity of the problem is mainly determined by two factors, the dimension of the subspace and the sparsity level of the permutation. For dense permutations (all coordinates are being permuted in an arbitrary manner), the problem is in principle extremely difficult. However, if the subspace dimension r is less or equal than 5, then the algebraic method AIEM is tractable (its complexity is exponential in r, so already for r=6, AIEM is not applicable). In the face experiment of Fig. 4 r=4, AIEM works as expected both for dense and sparse permutations. In fact, for r=4, AIEM is more efficient than Algorithm 2. On the other hand, Algorithm 2 is by design suitable for sparse permutations, empirically succeeding for up to 60%-70% permuted entries. Moreover, its complexity is $O(r^2 m^2)$, and thus it can deal with much higher ranks than AIEM. So indeed, it succeeds in Fig. 4 for sparse permutations (last two rows) but fails as expected for dense permutations (top two rows). We will clarify this even further in the paper.
>
> 3. “Can the authors provide some discussion on a few reasonable necessary or sufficient conditions for data matrices to be generic in applications?” The definition of generic in the paper amounts to the requirement that the matrix does not satisfy certain polynomial equations that depend only on the problem parameters m,n,r. As such, it is in fact difficult to find matrices that are not generic. In practice, to make an $m \times n$ rank-r matrix $X$ generic, it is enough to take a factorization $X=BC$ of $X$ with B $m \times r$ and C $r \times n$, and add a little bit white noise on all the entries of B and C. This will keep the rank equal to r, and it will not satisfy the equations that describe the non-generic matrices for the problem at hand with probability one. Thank you for this interesting question; we will add this explanation in the revision.
>
> 4. “Is it not true that powerful de-anonymization methodology can be used to violate the privacy of certain data releases? That sounds potentially harmful, so it might be useful to give it some thought and add a bit of discussion.” We are happy to comment on this more in the paper. We believe that de-anonymization techniques can be used with either positive or negative impacts. As the reviewer acutely points out, they can be used by hackers to violate privacy, which is certainly negative. However, they can also be used as testing tools in designing anonymization protocols that are difficult to break. In that regard, there is a positive message in our paper: arbitrary permutations are extremely difficult to recover - one would need to solve an intractable polynomial system. On the other hand, partial permutations are more susceptible to de-anonymization attacks and thus are not recommended.

---

> > ### Comment · Reviewer_Xvq6 · 2021-08-31
> > **Acknowledging rebuttal**
> >
> > Dear authors,
> >
> > Thank you for your clear responses to my questions. I am looking forward to reading the updated version of your paper containing this extra discussion.
> >
> > Best regards

---

### Official Review · Reviewer_azUW · 2021-07-24

**Rating:** 8
**Confidence:** 5

**Summary:**

The paper introduced a variant of PCA in the presence of noise, i.e., a variant of robust PCA. The "noise" in this case is permutations of the column entries of the data matrix. Using concept from Algebraic geometry, the authors showed that they can get the solution upto permutation of the rows. Experimental results have shown that the proposed method can recover the data matrix under such corruption.

**Limitations And Societal Impact:**

Yes

**Main Review:**

This is an excellent paper in terms of theoretical contribution. In my knowledge, this is the first paper which uses algebraic geometry to have a polynomial solution of robust PCA where the noise if permutation of column entries.

Strength:
(a) Theorem 3 is very interesting as it shows that the corrupted data matrix admits a partition into a inlier and outlier based on the ground truth column space. This is the main finding that helps to give the two stage proposed algorithm
(b) The polynomial two stage algorithm is natural as in the first stage the authors tried to find an estimation of column space. Given an estimation of the column space, using Theorem 3 the authors used a sensing method as second stage to get the partition between inliers and outliers.

My main criticism is readability of the paper, please see the comments below
(a) The paper is very dense in places and even with a fair knowledge about algebraic geometry the reader may find it difficult to read, e.g., (i) the authors should separate definition/discussion about Zariski topology in separate paragraph, line 115 - 132 can be  restructured (ii) Theorem 3 should be introduced before and theorem 1 and 2 may be discussed in detail in supp. material (iii) The discussion regarding polynomial root finding starting from line 150 is dense and can be restructured. (iv) The authors did not spend enough lines explaining the polynomial algorithm, for example complexity analysis of Algorithm 2 needs to be there.

(b) The experimental results should be cut down as the main contribution is Theorem 3 and the polynomial algorithm.

I am willing to strong accept after this restructuring for better readability.

**Time Spent Reviewing:**

6

---

> ### Author Response · Authors · 2021-08-08
> **Response to Reviewer azUW**
>
> We thank the reviewer for acknowledging the strength of the paper. The comments and suggestions on readability are well-taken, and we will follow them to our best. We wanted to make the paper solid on all aspects (theory, algorithms, experiments), and this came at the cost of dense writing at a few places. Please note that if the paper gets accepted, we will have an additional page, which will enable us to make the writing and structure, especially in section 2, more friendly. We will also discuss the complexity of Algorithm 2; this is O($m^2 r^2$); please see also our response to reviewer 9vKy about algorithm complexities.

---

### Decision · Program_Chairs · 2021-09-27

**Decision:**

Accept (Poster)

**Comment:**

The authors identified an interesting problem about principal component analysis with unknown permutations of the feature in each data sample. They then proposed an algorithm and provided performance guarantee. All 4 reviewers agreed to accept this paper.